# Possible factors determining global-scale patterns of crop yield sensitivity to drought

**Vempi Satriya Adi Hendrawan** [1,2]*, **Daisuke Komori**[2,3]*, **Wonsik Kim**[4]

**1** Department of Civil and Environmental Engineering, Universitas Gadjah Mada, Yogyakarta, Indonesia, **2** Graduate School of Environmental Studies, Tohoku University, Sendai, Japan, **3** Graduate School of Engineering, Tohoku University, Sendai, Japan, **4** Institute for Agro-Environmental Sciences, National Agriculture and Food Research Organization, Tsukuba, Japan

* vempi@ugm.ac.id (VSAH); daisuke.komori.e8@tohoku.ac.jp (DK)

**Data Availability Statement:** Datasets generated or analyzed during the current study are available at http://doi.org/10.6084/m9.figshare.21298785.

**Funding:** D.K. was partly supported by the International Joint Graduate Program in Resilience

## Abstract

In recent decades, droughts have critically limited crop production, inducing food system shocks regionally and globally. It was estimated that crop yield variability in around one-third to three-fourths of global harvested areas is explained significantly by drought, revealing the notable vulnerability of crop systems to such climate-related stressors. However, understanding the key factors determining the global pattern of crop yield sensitivity to drought is limited. Here, we investigate a wide range of physical and socioeconomic factors that may determine crop-drought vulnerability in terms of yield sensitivity to drought based on the Standardized Precipitation Index at 0.5˚ resolution from 1981 to 2016 using machine learning approaches. The results indicate that the spatial variations of the crop-drought sensitivity were mainly explained by environmental factors (i.e., annual precipitation, soil water-holding capacity, soil acidity, annual potential evapotranspiration) and crop management factors (i.e., fertilizer rate, growing season). Several factors might have a positive effect in mitigating crop-drought vulnerability, such as annual precipitation, soil water holding capacity, and fertilizer rate. This study quantitatively assesses the possible effect of various determinants which might control crop vulnerability to drought. This understanding may provide insights for further studies addressing better crop vulnerability measures under future drought stress.

## Introduction

The impact of global climate variabilities on crops has been well investigated. Empirically, around one-third of the variability of crop measures (i.e., crop yield, production) [1–3] and harvested area globally [4] are explained by climate variabilities during the last several decades. As one climate extreme, drought has been widely recognized as a notable limiting factor for crop yield and production [5, 6]. One-third to three-fourths of the global harvested areas experienced yield losses due to drought [7, 8], corresponding to approximately 166 billion US dollars from 1983 to 09 [7]. The serious drought impact on crops may unveil the global crop system's vulnerability to climate-related stressors, which can later threaten local or global food security in the current and future climate [3, 9, 10].

and Safety Studies, Tohoku University. The funder had no role in study design, data collection and analysis, decision to publish, or preparation of the manuscript.

**Competing interests:** The authors have declared that no competing interests exist.

Approaches for understanding the drought risk on the crop system may include assessing its potential impact and crop vulnerability under specific dry condition definitions (e.g., precipitation deficit, soil moisture stress) [11–13] during the growing period as a hazard and exposure measures [9]. The drought impact on crops can be measured by damages or losses on crop production measures (i.e., crop yield, crop production, or harvested area) due to certain drought conditions, whereas the vulnerability can be presented by the sensitivity or susceptibility of the crop production responding a drought severity [9, 14]. While the drought impact on crops and their vulnerability is growingly evident (through field-scale experiments to larger scale based on global physical modeling and statistics [1, 15–18]), quantitative identification of multiple factors that may determine such crop-drought relationship patterns across global agricultural systems is limited in previous works.

Agricultural risk and vulnerability to drought in a region may be determined by several factors [14, 17], which can be generally grouped as physical (e.g., climate, soil, topography, crop production factors) and socio-economic (e.g., crop management, agricultural input, field size, demography, farmer income, Gross Domestic Product (GDP) per capita, technology) [19–21]. These determinants govern the adaptive capacity of a system to climate disruptions over time. For instance, studies [7, 22] reported that technological improvements (i.e., high-yield breeding, adoption of drought-tolerant seed) associated with per capita GDP growth could further increase the resilience of crop production systems to droughts in a region. Current reports also revealed that more resilient agricultural systems, due to social, institutional, and agroecological factors [23] could adapt to stress and recover better than those less resilient, even with a comparable or higher climate extreme exposure [20, 22]. Therefore, quantifying the adverse impact on crops due to extreme events such as a drought is imperative to understanding crop vulnerability to climate disruptions since drought impact is known to be one of the most disastrous climate extremes causing crop damage.

Furthermore, how socio-economic factors drive the global crop-drought relationship is often overlooked. Several studies [12, 19, 21] attempted to quantitatively explore how various physical and socio-economic factors might control crop productivity measures (i.e., crop yield, production). Even though numerous existing local studies with a specific context and underlying physical and socio-economic conditions existed [24, 25], these studies might not represent the understanding to a larger extent. On a global scale, previous studies often use aggregated country or sub-country level data to assess socio-economic factors' influence to crop vulnerability [19, 20, 26, 27]. This approach might not represent heterogeneous agricultural systems, given that adaptive capacity may differ within local or smaller levels [28–30].

For instance, existing global scale studies by Simelton et al. [20] and currently Kinnunen et al. [31] have attempted to understand the socioeconomic factors which determine crop vulnerability to climatic stress (i.e., drought and heat). They employed several socioeconomic proxies at subnational and national levels, such as governance, GDP per capita, human development index, fertilizer, water stress index, and irrigation infrastructure indicator. They emphasized the importance of socioeconomic factors (e.g., income level, governance, fertilizer application) in defining global crop vulnerability to climatic stress. These existing studies also suggested crop production in countries with higher agricultural investments (i.e., higher fertilizer application) is generally less vulnerable to drought, despite a variation across crops and the type of regions. However, some complex and nonlinear relationship has been reported, particularly for economic indicator (e.g., GDP per capita, human development index) [20]. Moreover, these previous studies have not discussed the effect of importance and the direction of how a wide range of factors combining physical (e.g., climate, soil, topography, irrigation, crop production factors) and socioeconomic variables affect crop-drought vulnerability in a finer spatial scale (i.e., grid-scale).

Therefore, this study addresses the above gaps of lacking quantitative evidence of the effect of multiple factors (i.e., physical, socioeconomic) in determining global crop-drought vulnerability based on gridded historical data. Particularly, this present study explores 1) how much global harvested area and crop yield anomaly are significantly affected by drought and 2) how the key factors mitigate or exacerbate the crop-drought vulnerability in terms of their crop yield sensitivity to drought. Here, we examine the sensitivity of maize, rice, soybean, and wheat yield to drought based on a meteorological drought indicator as drought proxy (Standardized Precipitation Index, SPI) with a spatial resolution of 0.5˚ grid-cell from 1981 to 2016. Finally, we explore how the key factors determine global crop-drought sensitivity using machine learning approaches. These study results are necessary to indicate which factors are important to determine the global crop sensitivity to drought on a global scale. This may provide light for further attempts to understand crop system vulnerability and its determinants for future adaptation and mitigation efforts in the global context.

## Materials

The dataset used in this study mainly consists of global gridded crop yield data, the Standardized Precipitation Index (SPI), and several indicators that possibly control crop sensitivity to drought. We consider all the data for 1981–2016 associated with this study analysis period. Since the spatial resolution of the datasets varies, we re-gridded all the datasets to 0.5˚ using bilinear interpolation.

## Drought indicators

Here, the drought indicator is based on one of the meteorological drought indices, the Standardized Precipitation Index (SPI) [32]. While SPI is solely based on precipitation, this index has been widely used and well-recognized in previous drought studies and applications [33–36]. Other drought indexes such as the Palmer Drought Severity Index (PDSI), the Standardized Precipitation Evapotranspiration Index (SPEI), and other indices (e.g., soil moisture anomaly) have also been used but require more parameterization, which may introduce additional uncertainties [37, 38]. Therefore, the simple but robust drought definition based on SPI is employed in this study to reveal the global drought pattern from 1981 to 2016.

SPI is obtained based on the standardization of monthly precipitation accumulation within various timescales (e.g., 3, 6, 9, 12 months) over the window period of 1981–2016; thus, the standardized value finally has a mean of zero and a standard deviation of one. Before standardization based on the normal Gaussian distribution, the data is fitted based on the Gamma distribution [39]. Here, SPI is developed based on several historical global gridded precipitations to obtain the best agreement among datasets according to their ensemble mean (GPCC, CRU, PRECL, UDEL, CPC, MSWEP, MERRA-2, and ERA-5 dataset) [8]. SPI is produced for each dataset, and then the mean of the eight selected datasets is obtained to represent ensemble SPI as the drought index.

The drought index (*DI*) based on negative SPI is used to represent the inter-annual drought condition in this study (**Eq 1**) [8]. We obtain yearly *DI* based on monthly SPI on harvest month each year. The harvest month data for each grid cell is obtained from the crop calendar dataset during the year 2000 [40] and is constantly applied every year from 1981 to 2016. The drought index (*DI*) is calculated as:

$$DI_t = \begin{cases} -SPI_{m,t}, & SPI_{m,t} < 0 \\ null, & SPI_{m,t} \geq 0 \end{cases} \tag{1}$$

where $SPI_{m,t}$ is the 9-month SPI (SPI-9) in harvest month $m$ and year $t$. SPI with a 9-month timescale is used to indicate a long-term precipitation anomaly which may indicate a more prominent drought [32, 33]. The spatial variation of the global crop-drought relationship is broadly similar when different SPI timescales are used [8].

### Crop yield anomaly

The percentage of yield anomaly ($\Delta Y$) is estimated based on the gridded dataset of historical yield (GDHY) [41] for the major crops: maize, rice, soybean, and wheat (**Eq 2**). We calculate long-term crop yield trends as normal or expected yields ($\bar{Y}$) using the local polynomial regression method [8]. This method is selected since it can account for possible nonlinear trends and works well for limited data series, particularly compared to the moving average method [42]. This long-term trend generally represents the improvement of technological advances in producing higher yields in a region, while the yield anomaly subtracted from the expected yield is assumed to be caused by climate-related factors, although other affecting factors, such as socioeconomic factors, may also influence this short-term anomaly.

$$\Delta Y_t = \frac{Y_t - \bar{Y}_t}{\bar{Y}_t} \times 100\% \tag{2}$$

where $Y$ is crop yield (t ha$^{-1}$) and $\bar{Y}$ is the trend obtained by local polynomial regression (t ha$^{-1}$) in a given year $t$.

### Crop-drought determinants

We obtain several possible factors determining crop yield and drought relationship. We define the possible determinants as (1) climate factors [1], (2) terrain factors [43, 44], (3) soil factors [45, 46], (4) irrigation factors [26, 27], (5) crop production (hereafter production) factors [47], (6) fertilizer factors [20, 48], and (7) socio-economic factors [12] (see **Table 1**). All variables shown in **Table 1** are then used as independent variables to train the Random Forest model (see Methods). Moreover, it is noteworthy that we use an average value or a subset of the dataset in a specific year between 1981–2016, despite a possible change during the study period. Previous studies [46] have also used this approach, and year-specific datasets are widely employed, especially in the case of crop calendar and harvest area datasets [49, 50], which were only available around the year 2000.

## Methods

### Crop-drought sensitivity

We estimate crop-drought sensitivity as the slope coefficient ($\beta$) of a linear regression between yield anomaly ($\Delta Y$) and $DI$ in the given year as time series (**Eq 3**). Statistical significance considered in this study is based on a two-tailed t-test with a threshold of $P < 0.05$.

$$\Delta Y = a + \beta \cdot DI \tag{3}$$

where $a$ and $\beta$ are the model intercept and slope, respectively. These linear regression parameters are fitted based on the ordinary least-squares method using the "statsmodels" library in Python [60] (see **S1 Fig** for an example of the regression model in a grid cell). This calculation is applied for each grid cell and crop. Here, crop-drought sensitivity refers to the simple linear relationship, despite possible nonlinear responses and more complex relationships occurring across regions. Nevertheless, this simple model allows us to understand the general pattern of how crop yield responds to drought, as demonstrated by previous studies [1, 61].

**Table 1. Determinants used to describe the crop-drought sensitivity analyzed in the study.** The dataset sources and format are listed in S1 Table.

| Class | Variable | Unit | Period | Source |
|---|---|---|---|---|
| Climate | Mean annual precipitation | mm year$^{-1}$ | 1981–2016 | [51] |
| | Mean annual PET | mm year$^{-1}$ | | |
| | Mean temperature | ˚C | | |
| Terrain | Elevation | m | 2000 | [52] |
| | Slope | - | | |
| Soil | Topsoil saturated hydraulic conductivity [a] | cm day$^{-1}$ | 2013 | [53] |
| | Topsoil clay amount | % weight | 2000 | [54] |
| | Topsoil organic carbon | kg C m$^{-2}$ | | |
| | Topsoil acidity | pH | | |
| | Water-holding capacity | mm | 1993 | [55] |
| Irrigation | Area equipped for irrigation (AEI) | % of land area | 2005 | [50, 56] |
| | Area actually irrigated (AAI) | % of AEI | | |
| | Area irrigated with groundwater (AEIGW) | % of AEI | | |
| | Area irrigated with water from non-conventional sources (AEINC) | % of AEI | | |
| Production | Growing season length | days | The 1990s –early 2000s | [40] |
| | Harvested area | km$^2$ year$^{-1}$ | 1998–2002 | [49] |
| Fertilizer | Nitrogen rate application | kg ha$^{-1}$ | 2000 | [57] |
| | Phosphorus rate application | kg ha$^{-1}$ | | |
| | Potassium rate application | kg ha$^{-1}$ | | |
| Socioeconomic | Gross Domestic Product (GDP) | billion United States Dollar (USD) | 2015 | [58] |
| | GDP per capita | USD | | |
| | Population density | people km$^{-2}$ | 2015 | [59] |

[a]30cm depth

## Random forest

Here we use one of the machine learning algorithms, Random Forest [62], to reveal the key factors defining the crop-drought relationship. We use Random Forest algorithm due to its relatively better accuracy, interpretability, and efficiency with large training datasets than other machine learning methods (e.g., decision trees, nearest neighbors) [62–64]. In addition, we test the sensitivity of our results by comparing the model performances estimating crop-drought sensitivity with the outputs of two other machine learning algorithms: support vector regression (SVR) [65] and gradient boosting (XGBoost) [66], implemented in the R library, "caret" [67].

We set the crop-drought sensitivity (*β*) as the response variable and various possible determinants (**Table 1**) as independent variables or predictors. The Random Forest model has been used to detect a nonlinear climate-crop system relationship [2, 46]. Here, the Random Forest algorithm is applied using the "randomForest" library in R [68]. We use parameter input as default settings (e.g., number of trees = 500) since the model skill with basic setting is adequate to reveal the variable importance and their influence on global crop sensitivity [46]. The model is fitted with the data from all available grid cells (representing spatial dimension) for each crop. During the calculation process, explained variance is calculated based on OOB (out-of-bag) error estimation by subsampling with replacement in the training samples [68].

Further, we obtain relative importance [66] for each variable based on the fitted models. Here we use the metric measuring number of tree nodes employing the variable to reduce the model's impurity averaged across all trees [68]. The "randomForest" calculates and scales it as

the total importance for all variables equals one. Then, for ease of comparison, we normalize them into the 0–1 range, representing the minimum and maximum values, respectively. Finally, we plot all models based on different settings and rank them from the most important parameter to the least.

We use a partial dependence plot from the fitted models to understand each variable's relationship with crop-drought sensitivity as the response variable [66]. The partial dependence allows selecting small subsets or single variables to reveal a relationship between predictor(s) and a response variable, similar to sensitivity analysis [2]. Here, we pick up the six most influential variables based on their overall relative importance (from the average of all experiment setups) to demonstrate their relationship with crop-drought sensitivity. This enables us to estimate the effect of the important variable to determine global crop sensitivity to drought. This study also calculates the variable importance and partial dependence plots based on SVR and XGBoost algorithms using "caret" and "pdp" R library [69] to investigate the robustness of our results.

## Results

### Crop-drought sensitivity pattern

We obtain crop-drought sensitivity based on the slope of the linear regression ($\beta$) between crop yield anomalies and the meteorological Drought Index (*DI*) based on SPI with a 9-month time-scale (**Fig 1**). Significant relationships are evident in around a quarter of croplands or over around

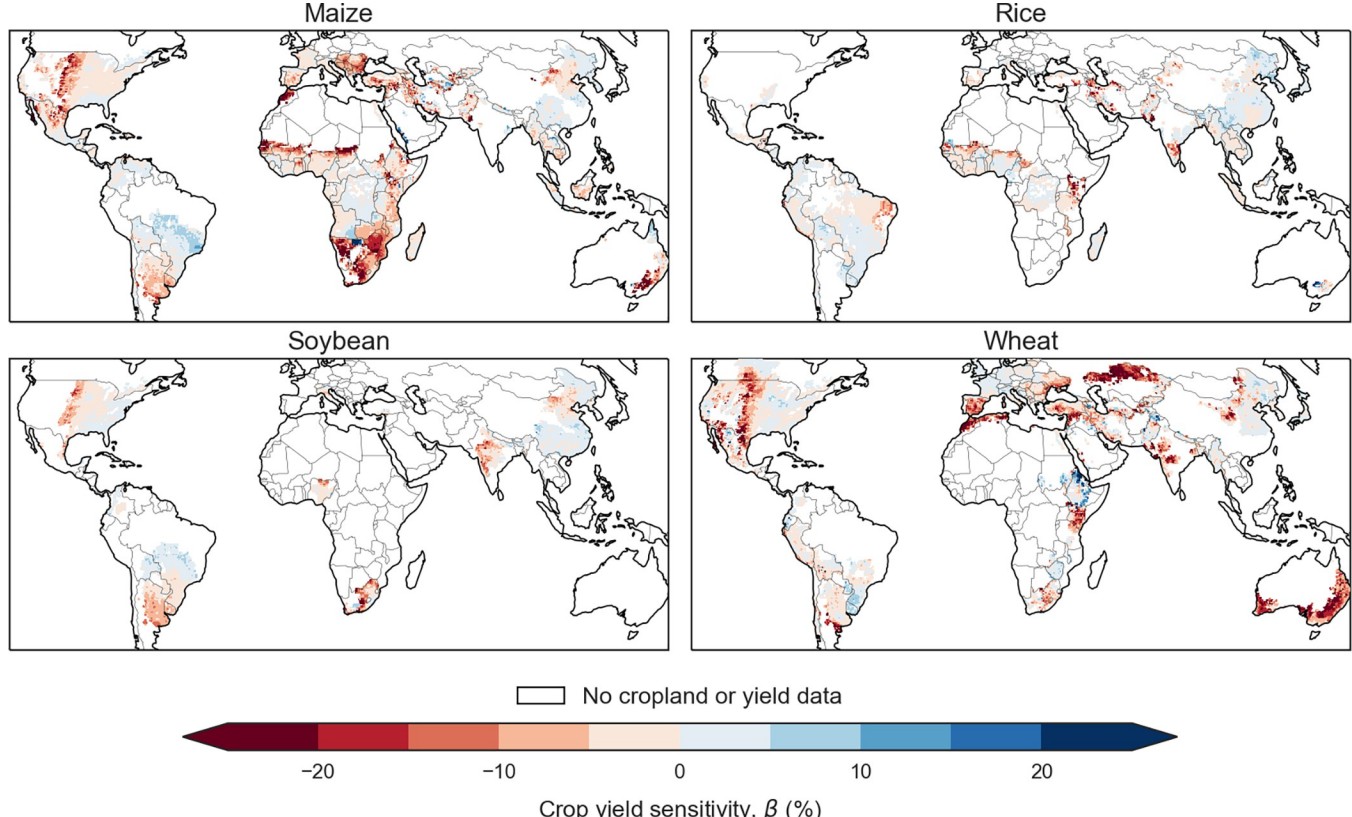

**Fig 1. Crop yield sensitivity to drought based on 9-month SPI across global croplands based on GDHY for maize, rice, soybean, and wheat.** Sensitivity is calculated by the linear slope coefficient ($\beta$) as crop yield anomaly (%) per drought magnitude based on SPI, while white cells show no cropland or crop yield data. We show a map over entire global croplands for a spatially broader interpretation. The map showing only areas with significant crop yield sensitivity to drought is provided in **S2 Fig** ($P < 0.05$). The map is made with the "cartopy" library in Python (version 0.17) and the Natural Earth free vector and raster map data (naturalearthdata.com), which are freely available through the public domain [70].

one-fourth of global maize, wheat, and soybean croplands, respectively ($P < 0.05$). Almost entire significant croplands show a negative correlation (~96%), confirming the damaging impact of drought on major crops. In contrast, the negative impact of drought on rice seems offset by yield benefits and remains low globally, with only 7% significantly correlated croplands [7].

Globally, drought reduces yield by 4%, 1%, 3%, and 6% for maize, rice, soybean, and wheat, respectively, with increasing drought index *DI* (i.e., precipitation deviations from normal condition per precipitation standard deviations). Over a quarter of the total croplands with significant drought impact, yields drop by around 14% by one drought unit (maize, soybean, and wheat). Furthermore, we point out primary regions with a substantial drought impact, comprising generally dry croplands such as The Great Plains, Africa (southern Africa, The Sahel, and the Horn of Africa), Australia, northeast China, the Mediterranean basin, and Central Asia (**Fig 1**) [7, 8]. Droughts also slightly damage maize and wheat yield in wetter regions such as South America, The Pampas, and the African and Southeast Asian tropics. While droughts induce rice yield loss in parts of The Sahel region and Northeast China and slightly in central Asia, rice in the tropics broadly shows weak sensitivity to drought. The higher crop yield sensitivity over drylands may indicate their higher vulnerability along with higher drought hazard exposures.

## Possible driving factors

We investigate the importance of several possible determinants as independent variables to control crop-drought sensitivity. We train the Random Forest models for regression with predictors of 22 variables. Results show that the variable predictors can explain overall 43%, 36%, 66%, and 37% of crop-drought sensitivity variations ($R^2$) with root mean square error (RMSE) of 8.61%, 5.89%, 4.63%, 10% (crop yield anomaly per drought magnitude, β) for maize, rice, soybean, and wheat, respectively. The explanatory skills of the 22 variables are robust across different types of machine learning algorithms (SVR and XGBoost), showing similar model performances with a small standard deviation of $R^2$ and RMSE by 1% and 0.2%, respectively, across crop types and machine learning algorithms (S2 Table). Furthermore, in rice, where the models less explain the crop-drought sensitivity variation, there are likely other more complex effects of factors contributing to rice vulnerability to drought [1, 7].

Here, we derive each variable's relative contribution to the models. While the important parameters vary across crops, we average their relative importance to reveal overall important variables (**Fig 2**). It shows that mean annual precipitation, water-holding capacity, soil acidity, Potential Evapotranspiration (PET), elevation, and potassium application rate are among the most driving factors, revealing that climate, soil, terrain, and agricultural input factors play an important role in crop-drought sensitivity variations. In particular, for each crop, the result suggests that the annual precipitation pattern becomes important for all crops, while water-holding capacity, elevation, PET, acidity, and potassium rate are relevant for some crops (**Table 2**). Regarding management factors, groundwater irrigation potential and population density are slightly important for maize, soybean, and rice.

The importance of variables is generally consistent across the machine learning models, particularly the order of the most important variables (see S3 Fig). Note that the importance of variables from the three machine learning algorithms (Fig 2 and S3 Fig) may slightly vary in absolute value due to different methods implemented [67]. The variable importance and partial dependence plot of SVR and XGBoost are obtained using "caret" library.

## Impact behavior of each factor

We obtain a partial dependence plot for each variable to describe their functional relationship with crop-drought sensitivity patterns (Fig 3 and S4 Fig). The subsequent sections discuss the

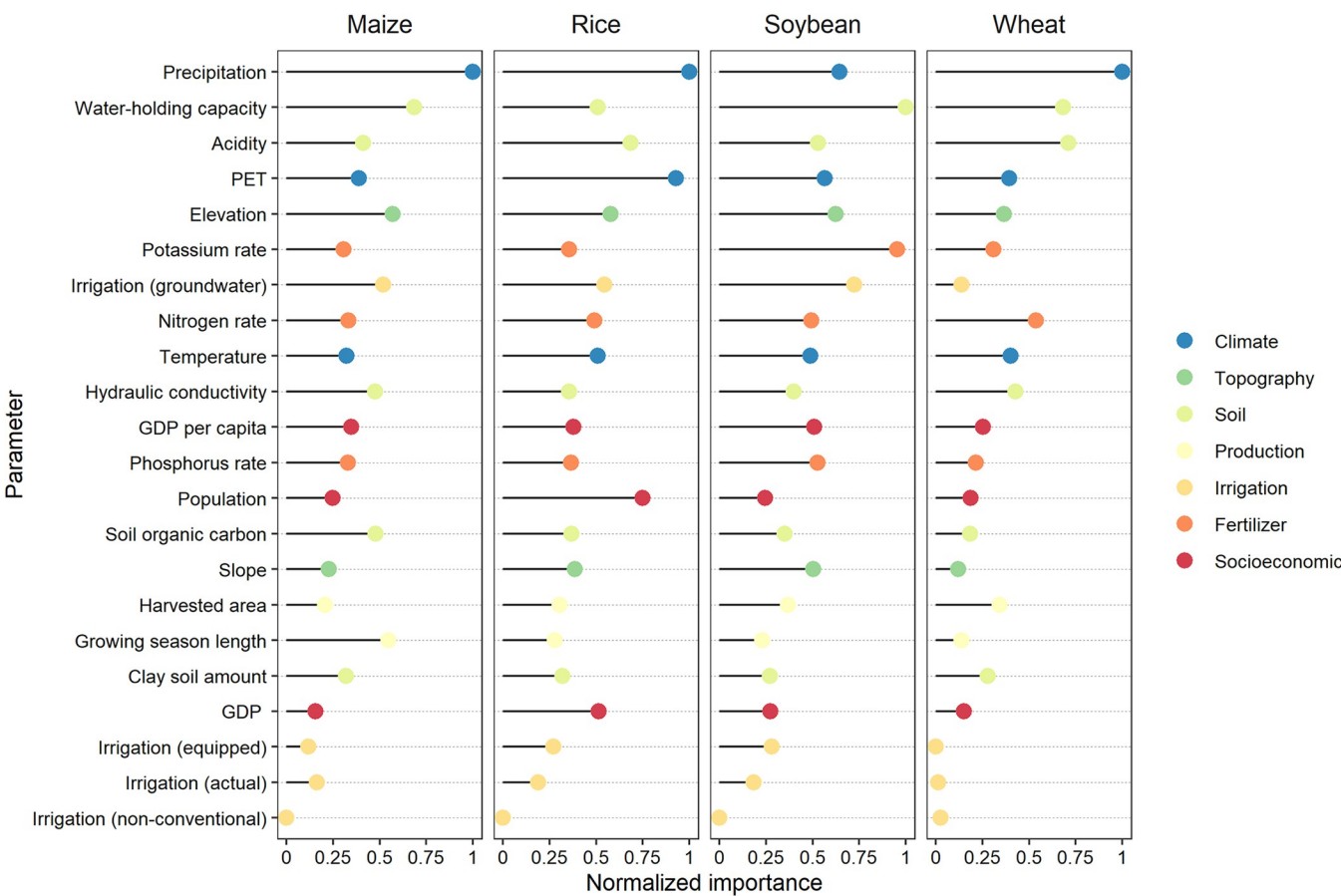

**Fig 2. The relative importance of the determinant factors in explaining crop-drought sensitivity based on random forest model.** The importance values are normalized to the 0–1 range computed separately across different crops. The colored points represent the relative importance of each variable. Label order of the y-axis indicates the order from the most important variables to the least based on the overall average across crops.

variable relationship for each parameter among different factors (climate, terrain, soil, production, irrigation, fertilizer, and socioeconomic factors) based on Random Forest model. We also obtain the partial dependence plots based on SVR and XGBoost (S5 and S6 Figs). The results confirm that most estimated relationships between various determinants and crop yield sensitivity are generally robust across the models.

## Climate factors

The most important factor, mean annual precipitation, demonstrates a clear relationship with crop-drought sensitivity; a higher precipitation rate alleviates crop yield reduction due to

**Table 2. Top six important parameters for each crop based on the average relative importance.** The overall average represents the average from combinations across crops corresponding to the order of the y-axis labels in Fig 2.

| Rank | Maize | Rice | Soybean | Wheat | Overall average |
|---|---|---|---|---|---|
| 1 | Precipitation | Precipitation | Water-holding capacity | Precipitation | Precipitation |
| 2 | Water-holding capacity | PET | Potassium rate | Acidity | Water-holding capacity |
| 3 | Elevation | Population density | Irrigation (groundwater) | Water-holding capacity | Acidity |
| 4 | Growing season length | Acidity | Precipitation | Nitrogen rate | PET |
| 5 | Irrigation (groundwater) | Elevation | Elevation | Hydraulic conductivity | Elevation |
| 6 | Soil organic carbon | Irrigation (groundwater) | PET | Temperature | Potassium rate |

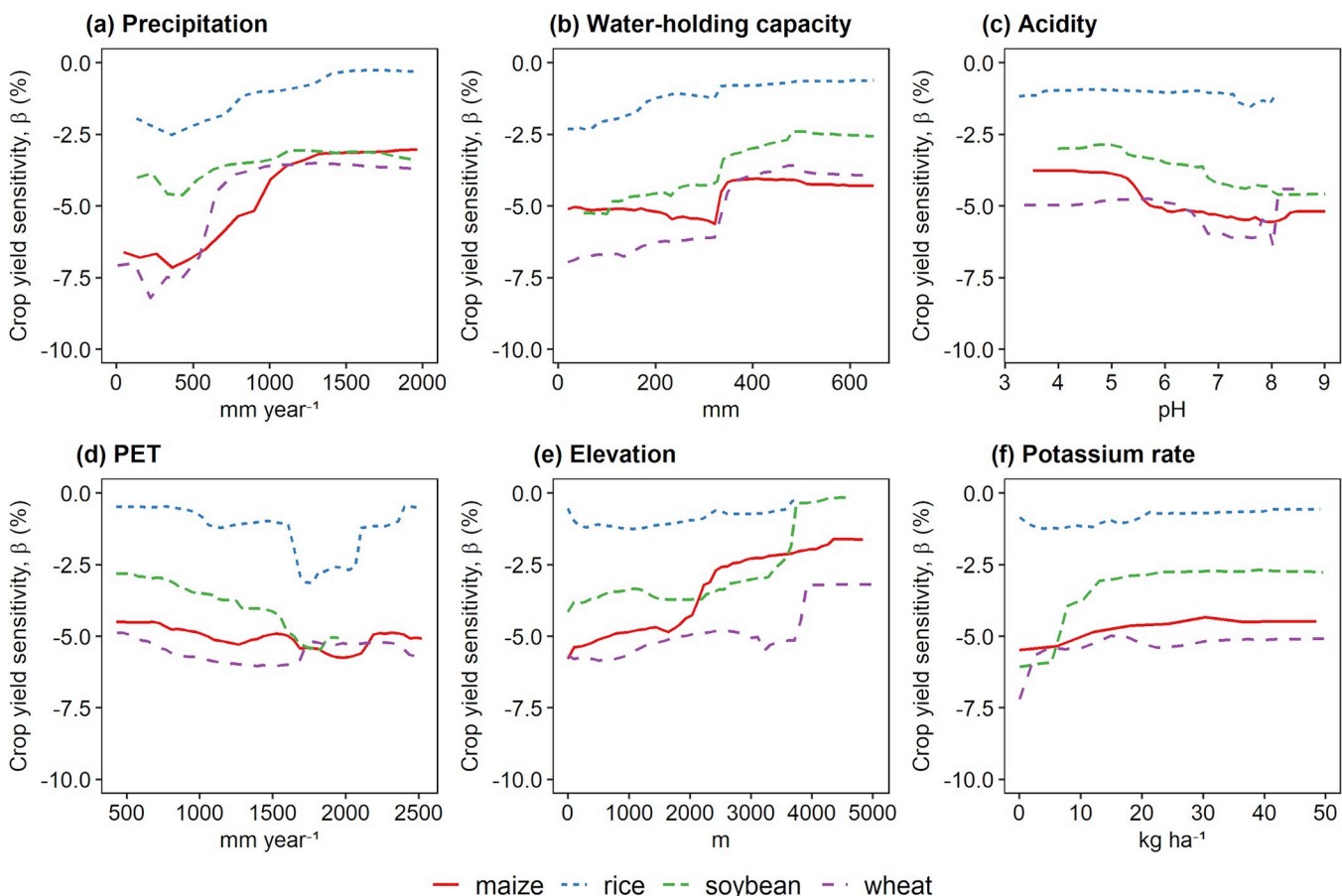

**Fig 3. Relationship between the response variable of crop-drought sensitivity ($\beta$) as the response variable and the six selected determinant factors as independent variables based on each partial dependence using random forest models.** The order shows higher to lower overall relative importance across crops (see **Table 2**). The lower negative y-axis indicates higher yield loss due to drought. The partial dependence plot for all variables (22 variables) is provided in **S4 Fig**.

drought. For instance, based on functional relationships in maize and wheat in which the yield loss sensitivity is high, the higher annual precipitation (e.g., ~500–1000 mm year$^{-1}$) increase may mitigate yield loss (**Fig 3**). This result indicates that crop yield in semi-arid and arid regions with less precipitation is more sensitive to drought impact, as reported elsewhere [7, 71–73]. Furthermore, this confirms that crop loss sensitivity increases in most drylands, where severe hazards and higher vulnerability occur concurrently. The other climate factor, PET, shows an inverse response to precipitation—higher PET tends to aggravate yield loss sensitivity due to drought, while temperature slightly determines the response variables.

## Terrain factors

The higher elevation is found to be an important parameter with a diminishing effect on yield loss sensitivity. The response is particularly evident for maize, soybean, and wheat. This implies that more serious crop-yield loss sensitivity due to drought is generally located in low-lying cropland; while in higher elevations, where the precipitation rate is generally higher, drought impact may be lessened [74]. Moreover, atmospheric vapor demand would generally be lower at higher altitudes, which may alleviate moisture stress due to drought [75]. On the other hand, the other terrain factor, the slope, does not show a clear relationship with the drought-induced yield sensitivity.

## Soil factors

The most important soil factor, water-holding capacity, generally shows a consistent pattern for all crops. The higher value may slightly moderate drought-induced yield loss, as indicated in the previous studies [76, 77]. For instance, this parameter's functional relationship shows a decrease in drought-induced yield loss in all crop regions by around 330 mm water-holding capacity. The other soil factor, soil acidity, tends to increase the drought-induced yield loss by pH values ranging from 7 to 8. In this range, soil condition is generally more saline, while soil may become alkaline with a pH of more than 8.5 [72]. With these conditions, salt and alkali stress may simultaneously restrict crop development in semi-arid and arid regions, further exacerbating crop yield loss to drought [78, 79]. Moreover, hydraulic conductivity may also lessen drought-induced yield loss, especially for maize and wheat within hydraulic conductivity values (i.e., up to 25 cm d$^{-1}$). This dependence generally may be related to more effective water transport and increasing soil water content potential as soil hydraulic conductivity increases [80, 81], despite more complex mechanisms that may be involved [15, 82]. The other parameters: soil organic carbon and clay soil amount, however, generally remain less influence given lower relative importance in the model, despite their importance in some cases (e.g., higher clay soil amount in wheat seems to slightly aggravate negative crop sensitivity to drought).

## Production factors

The crop production factor, growing season length, is revealed as one of the important factors that may worsen wheat yield loss due to longer season length, particularly for maize and wheat. The longer season length is generally attributable to winter wheat being dominated by a global rainfed system, and the crop may be highly reliant on precipitation and sensitive to precipitation variability. For instance, the longer maize growing season (>~130 days) may exacerbate the yield loss. This growing season covers most of the major maize belts in higher latitudes, including most semi-arid and arid lands, while the shorter season spans around the tropics. The other parameter, the harvested area, may exhibit lesser effects.

## Irrigation factors

Overall, irrigation parameters (i.e., percentage of area equipped for irrigation, AEI; actual irrigation, groundwater; and non-conventional water sources) show an unclear relationship with the yield sensitivity, despite a slightly higher relative importance, especially for the groundwater irrigation parameter. The remaining irrigation parameters tend to have less control over the drought-induced yield sensitivity. We acknowledge that these results contrast with previous studies arguing that irrigation may partly mitigate drought impact to crop yield.

## Fertilizer factors

The fertilizer factors, potassium and phosphorus application rates, are revealed among the important variables. In general, their higher application rate may slightly alleviate yield loss due to drought. These fertilizer applications may effectively relieve drought stress on crops and enhance drought tolerance [83–85]. This mechanism may be generally explained by reducing the uptake of toxic nutrients and strengthening the physiological efficiency [83], which may improve overall resilience to drought, while in the particular further mechanism of the role of nutrients to mitigate drought stress on plants has been discussed in previous studies [86]. Moreover, the other fertilizer factor, nitrogen application rate, does not show an apparent relationship with the yield sensitivity indicator, despite a notable exception for the wheat with an easing effect on the drought-induced yield loss.

### Socio-economic factors

GDP per capita relatively shows a stronger relationship with the yield sensitivity than the GDP parameter, especially in the case of wheat and soybean. However, these relationships remain weak, preventing us from drawing a conclusion. The other factor, the population indicator, does not determine yield sensitivity to drought.

## Discussion

The present study assesses crop sensitivity to drought from 1981 to 2016 based on a meteorological drought index using a 9-month Standardized Precipitation Index (SPI) and gridded crop yield datasets for maize, rice, soybean, and wheat. Our estimates reveal that over one-fourth of global cropland is significantly sensitive to inter-annual drought, which generally agrees with previous studies [2, 7, 73], despite slightly lower estimates than the previous study using a similar approach and dataset [7]. A negligible drought impact on rice is also confirmed in this study, which may be attributable to its low exposure to drought with general moisture surplus as reported elsewhere [61, 87]. Here, we highlight a corresponding result pattern among related studies based on specific drought indicators and crop statistics during historical periods [7, 88, 89]. The major global drylands are the hotspots of the drought-sensitive regions, particularly for the three most affected crops: maize, soybean, and wheat. The significance of the regions is also widely highlighted by previous studies [7, 11, 73].

This study explores several possible factors determining crop sensitivity to drought. We indicate that environmental factors (i.e., climate, soil, terrain) have an important role in mitigating crop yield loss due to drought, while management factors such as fertilizer, growing season, and GDP per capita show moderate influence, while the irrigation factor seems to have lesser control, in contrast to their typical role in mitigating the impact of drought [2, 27, 61, 87, 90]. This result may suggest that drought events in our scale are too large to prevent crop losses by irrigation; water deficits occurred extensively in the case of large-scale droughts. Previous study by Kinnunen et al. [31] also indicated the importance of fertilizer application to alleviate crop yield variation during dry and hot years, while the irrigation factor was also relatively less important due to possible differences between actual conditions and data representation.

We further discuss these findings regarding the irrigation effect as follows. First, we may not detect the significant importance of irrigation factors as we cannot independently disentangle the impact of drought on irrigated and rainfed crop yields. Here, we use mean annual yield involving multiple crop growing systems (irrigated and rainfed) and seasons (major and minor) [41]. Therefore, our estimates may offset the mitigating effect of irrigation due to a comparable or even larger negative impact of drought on rainfed crops within the same grid location. Previous studies primarily conducted on a local or regional scale employed irrigated and rainfed yield data separately to obtain drought impact on each system [90], while further challenges related to data availability may arise when dealing with a global scale analysis.

On the other hand, another study [2] detected response of inter-annual yield using sub-country-level data (combined rainfed and irrigated yields) on each classified irrigated and rainfed cropland based on a specific definition (e.g., irrigated cropland is defined as >80% of harvested area irrigated and otherwise is rainfed). Similarly, we classify irrigated and rainfed cropland based on the similar global harvested area dataset [49] and the classification used in the previous study [2]. We then calculate the share of drought-affected areas for each crop-drought sensitivity bin (**Fig 4**). We find that the rainfed system considerably dominates the drought-affected regions resulted in this study, especially for soybean (99% of the total calculated harvested area), maize (89%), and wheat (83%), while in the case of rice, the proportion of irrigated cropland is relatively higher (63%). Irrigation tends to mitigate rice yield loss as the

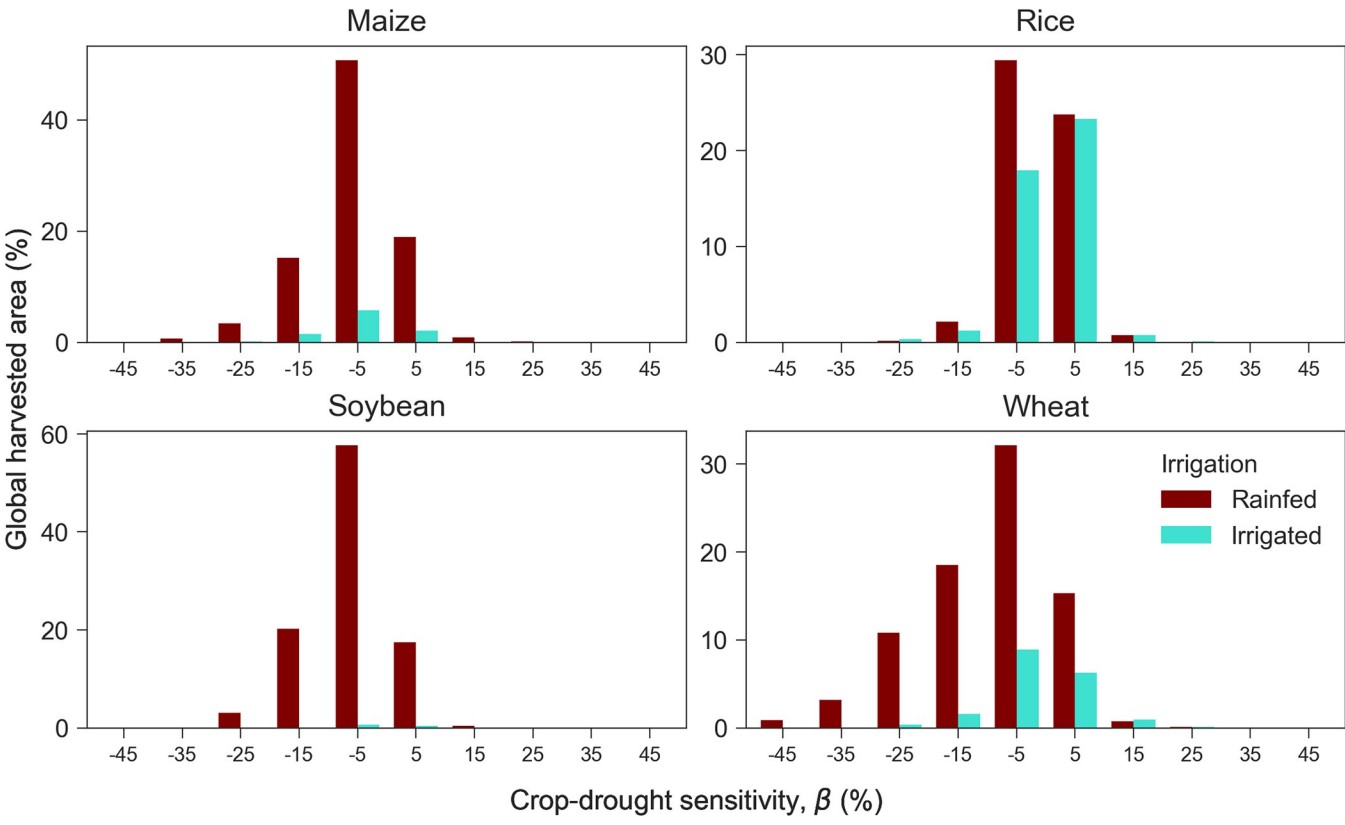

**Fig 4. Crop-drought sensitivity in each rainfed and irrigated defined by >80% of irrigated harvested area for maize, rice, soybean, and wheat.** The x-axis represents the bins of crop yield sensitivity, and the y-axis represents the percentage of the harvested area where data is available. The dashed line indicates the center value (zero), separating yield loss (negative) and yield gain (positive).

centroid of the bins tends to be positive (yield gain), while rainfed rice tends to have yield loss. For the major rainfed crops (maize, soybean, wheat), there seems to be no considerable difference between irrigated and rainfed system responses; both exhibit mainly yield loss despite a large difference in cropland share. Nevertheless, the vast rainfed system is negatively impacted by drought, shown by an extensive area proportion attributable to yield loss, which indicates the substantial drought impact on this cropping system globally. Particularly, maize, soybean, and wheat are damaged by drought, equal to 65%, 69%, and 62% of the calculated harvested area. Therefore, we may argue that variability of irrigation fraction may not be adequate to explain crop yield sensitivity to drought owing to the dominance of a rainfed system over global croplands calculated in this study.

Further study may consider other important parameters for drought parameterization, such as temperature or evapotranspiration (ET), as they play a significant role in exacerbating drought impact through increasing evaporative demand [61, 91, 92]. The drought indicator can also be represented by other drought parameters, such as growing season soil moisture conditions based on physical modeling or remote sensing estimates on a global scale [37, 93]. Previous studies reported that the impact of drought on agricultural production due to lacking soil moisture is often associated with extreme heat, posing a more severe risk to crops [61, 94, 95]. Joint drought and heat events indicators may provide insight into these relationships [88, 96, 97]. We suggest considering these possible mechanisms accounting for hydrological features on drought estimates (rather than only meteorological factors, e.g., precipitation), especially in some regions agricultural systems impose more complex atmospheric soil interaction

and are heavily reliant on the other common water sources such as irrigation and groundwater [61, 98].

Moreover, we indicate that the overall effect of socio-economic factors, i.e., GDP per capita, is relatively weak in determining crop-drought sensitivity—which may be underestimated in this study. Specifically for soybean, higher GDP may slightly induce higher yield loss sensitivity to drought, while maize slightly responds differently. Previous studies [20, 99] reported that higher GDP per capita generally reflects larger average farm sizes [100], which are typically market-oriented. Furthermore, a previous study [99] suggested that this type of system may be more sensitive to drought when they tend to achieve maximum yield potential associated with higher risk instead of adjusting low-risk management practices during climate stress. Possible reasons are highlighted: when GDP per capita increases, conventional drought coping strategies may be reduced, while in poorer regions, traditional and well-settled adaptation strategies may be more preserved, possibly reducing the harvested yield volatility [20, 99]. However, this socio-economic indicator effect in defining the crop sensitivity to drought may remain complex in actual conditions. In contrast, numerous pieces of literature have indicated that the importance of GDP or capital indicators may decrease vulnerability (i.e., increasing adaptive capacity) due to more financial reserves for investing in, e.g., drought-tolerant breeding, irrigation facility, mechanization, or weather forecasting [13, 20, 25, 26]. Other studies also highlighted that crop resilience to drought is higher in rich and developing countries than in middle-income countries [20]. It is worth noting that this present study only considers a specific vulnerability measure (i.e., crop yield sensitivity, yield loss) based on the relationship between crop yield and meteorological drought. In contrast, a vulnerability in its comprehensive definition may consider more factors (i.e., drought-induced production and economic loss, food trade) [7, 9].

Finally, we acknowledge the limitation of this study related to the datasets used, particularly the reliance on the gridded dataset of historical yield (GDHY) and SPI-based drought indicator as the two main input parameters [8]. The crop-drought sensitivity patterns are also subject to change if other datasets (i.e., sub-country crop yield data [3]) and other drought indicators are used [37]. The drought definition used in this study can be further improved by considering other parameters (i.e., temperature, ET, soil moisture). Therefore, we acknowledge that crop yield sensitivity here is exclusively determined by the specific datasets used here, mainly obtained from the SPI-based drought indicator and yield dataset from GDHY [41]. It is noteworthy that the result of key determinants of the global pattern of crop sensitivity to drought may be sensitive to the data and model selection in this study.

At this point, challenges and questions remain for empirically understanding the complete drought characterization on global agriculture. Nevertheless, this study is worth undertaking since this provides the first attempt to broadly link several possible determinants with the crop-drought relationship to reveal the most influential parameter in defining the relationship patterns over the global scale. Overall, this study's results may provide a basis for measures to achieve food security against the pressures such as increasing severity and frequency of extreme events under climate changes, food demand due to the growing global population, and land availability [5, 13, 101].

## Conclusion

This paper explores key factors determining crop-drought sensitivity for maize, rice, soybean, and wheat for 1981–2016. Results suggest that maize, soybean, and wheat yield are significantly affected by drought in major dry croplands where they are mostly cultivated (e.g., The Great Plains, Africa, Australia, the Mediterranean basin, and Central Asia), while rice is generally

less affected. We link the crop-drought sensitivity with various determinants using the Random Forest model. Results reveal that environmental factors (i.e., annual precipitation, soil water-holding capacity, soil acidity, annual potential evapotranspiration) and management factors (i.e., fertilizers application rate, growing season) are among the key factors in controlling crop sensitivity to drought. Our study also reveals how such important factors affect crop-drought vulnerability: 1) crop yield in semi-arid and arid regions with low annual precipitation and higher PET is more sensitive to drought impact; 2) other important factors such as higher fertilizer rate, elevation, soil water holding capacity and shorter growing season have a moderate association with lower vulnerability, 3) the effect of other socioeconomic indicators (i.e., GDP per capita, irrigation) may be underestimated in our models generally due to limitation in modeling the actual mechanism by the selected proxies based the currently available global dataset. These results may improve our understanding of global crop-drought sensitivity patterns and their key determinants. Future studies are expected to address the remaining gaps demonstrated in this study to understand crop system vulnerability to drought toward future challenges to achieve food security.

## Supporting information

**S1 Table. List of dataset sources.** All datasets are freely available via websites.
(PDF)

**S2 Table. Output performance ($R^2$ and RMSE) of each machine learning algorithm trained in this study.** The support vector machine model is built using ´´´svmRadial´´´ based on the "caret" R library with the parameter sigma value (σ) of 0.1 and cost (C) of 1. Extreme gradient boosting is developed using "xgbTree" method in "caret" using its default fitting parameters (e.g., number of rounds of boosting = 500, maximum depth of a tree = 6, learning rate = 0.3). The values of the performance skills are based on the mean value from 10-fold cross-validation, except in the case of random Forest, which is solely based on OOB (out-of-bag) validation in this study. The unit percentage for RMSE denotes crop yield anomaly per drought magnitude (β).
(PDF)

**S1 Fig. Example of the linear regression between yield anomaly (Δ*Y*) and *DI* in a grid cell.** Crop drought sensitivity (β) is obtained based on the slope coefficient.
(TIF)

**S2 Fig. Crop yield sensitivity to drought based on 9-month SPI across global croplands based on GDHY for maize, rice, soybean, and wheat.** Sensitivity is calculated by the linear slope coefficient (β) with crop yield anomaly (%) per drought magnitude. White grid cells show no cropland or crop yield data, and grey shows the non-significant grid cells ($P \geq 0.05$).
(TIF)

**S3 Fig. The relative importance of the determinant factors based on Support Vector Regression (SVR) and Extreme Gradient Boosting models (XGBoost).** The importance values are normalized to the 0–1 range computed separately across different models. The colored bars represent the average relative importance across different algorithms. Label order of the y-axis indicates the order from the most important variables to the least based on the overall average across crops and models.
(TIF)

**S4 Fig. Relationship between crop-drought sensitivity (β) as the response variable and the determinants factors as independent variables based on each partial dependence using**

**Random Forest model.** The order shows high to lower average relative importance across crops based on this specific model (not overall). The lower negative y-axis indicates higher yield loss due to drought.
(TIF)

**S5 Fig. Same as S4 Fig but for SVR model.**
(TIF)

**S6 Fig. Same as S4 Fig but for XGBoost model.**
(TIF)

## Author Contributions

**Conceptualization:** Daisuke Komori, Wonsik Kim.

**Data curation:** Vempi Satriya Adi Hendrawan.

**Formal analysis:** Vempi Satriya Adi Hendrawan.

**Funding acquisition:** Daisuke Komori.

**Investigation:** Vempi Satriya Adi Hendrawan.

**Methodology:** Vempi Satriya Adi Hendrawan, Daisuke Komori.

**Project administration:** Daisuke Komori.

**Resources:** Daisuke Komori.

**Software:** Vempi Satriya Adi Hendrawan.

**Supervision:** Daisuke Komori, Wonsik Kim.

**Validation:** Daisuke Komori, Wonsik Kim.

**Visualization:** Vempi Satriya Adi Hendrawan.

**Writing – original draft:** Vempi Satriya Adi Hendrawan.

**Writing – review & editing:** Daisuke Komori, Wonsik Kim.

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
