## [Decision Letter · Decision Letter 0]

6 Oct 2022

PONE-D-22-24361Possible factors determining spatial patterns of drought impact on major crop yield sensitivity on a global scalePLOS ONE

Dear Dr. Hendrawan

Thank you for submitting your manuscript to PLOS ONE. After careful consideration, we feel that it has merit but does not fully meet PLOS ONE’s publication criteria as it currently stands. Therefore, we invite you to submit a revised version of the manuscript that addresses the points raised during the review process.

We look forward to receiving your revised manuscript.

Kind regards,

Arun Kumar Shanker

Academic Editor

PLOS ONE

Journal Requirements:

3. We note that Figure 1 and S2 in your submission contain [map/satellite] images which may be copyrighted. All PLOS content is published under the Creative Commons Attribution License (CC BY 4.0), which means that the manuscript, images, and Supporting Information files will be freely available online, and any third party is permitted to access, download, copy, distribute, and use these materials in any way, even commercially, with proper attribution. For these reasons, we cannot publish previously copyrighted maps or satellite images created using proprietary data, such as Google software (Google Maps, Street View, and Earth). For more information, see our copyright guidelines: http://journals.plos.org/plosone/s/licenses-and-copyright.

a. You may seek permission from the original copyright holder of Figure 1 and S2 to publish the content specifically under the CC BY 4.0 license.  

Reviewers' comments:

Reviewer's Responses to Questions

**Comments to the Author**

1. Is the manuscript technically sound, and do the data support the conclusions?

Reviewer #1: Yes

Reviewer #2: Yes

2. Has the statistical analysis been performed appropriately and rigorously? 

Reviewer #1: Yes

Reviewer #2: Yes

3. Have the authors made all data underlying the findings in their manuscript fully available?

Reviewer #1: Yes

Reviewer #2: No

4. Is the manuscript presented in an intelligible fashion and written in standard English?

Reviewer #1: Yes

Reviewer #2: Yes

5. Review Comments to the Author

Reviewer #1: 1. Is the manuscript technically sound, and do the data support the conclusions?

Yes, the manuscript is sound and data supports the conclusions. Further data on heat, soil salinity, water quality, cultivar, and other variables will further fine-tune the model.

2. Has the statistical analysis been performed appropriately and rigorously?

The manuscript is a model analysis based on existing data

3. The data is made available, English is good, and suitable for publicaiton

NOTE:

All comments are marked in the attached file. Addressing the comments as well as accepting the editorial changes is part of this review process and should be considered prior to publication.

Reviewer #2: The submitted manuscript investigated possible factors determining spatial pattern of drought impact on major crop yield sensitivity on a global scale

The overall framework of the study was well planned with enough information about the subject matter. Here are some specific comments with the hope that will help to clarify some things and improve the manuscript overall.

1. The authors did not express their aim well enough in the last paragraph of the introduction section and should be written more clearly about how the results of the study will contribute to the literature.

2. Should be added to more literature related to this study in the introduction section.

3. Line 182 – line 183 and Line 196 – line 201 Authors should revise the results section. There seems to be discussing rather than pointing out the result of the study.

4. Line 335 – line 347 Authors should rewrite and present them as results.

5. Authors should be more detailed about the data analysis session. The tool used for the analysis should be stated clearly with references.

6. Authors should make the results session come out clear and on point.

7. Authors should improve the discussion section and avoid repetition of results in the discussion section. More literature should be included to make it solid.

8. Conclusion section should be shortened and concrete.

9. The references section should be double-checked for the journal formatting style and rules.

6. PLOS authors have the option to publish the peer review history of their article (what does this mean?). If published, this will include your full peer review and any attached files.

Reviewer #1: No

Reviewer #2: **Yes: **Emmanuel Amponsah Adjei

---

## [Author Response · Author response to Decision Letter 0]

13 Oct 2022

Dear respected reviewers,

Thank you for allowing us to improve our manuscript for publication in PLOS ONE. 

We appreciate the time and effort the reviewers dedicated to providing feedback on our manuscript. 

In this revision, we have addressed all comments and suggestions.

We believe that our current manuscript is now ready for publication.

Sincerely

Vempi Satriya Adi Hendrawan

on behalf of the authors

---

## [Decision Letter · Decision Letter 1]

8 Nov 2022

PONE-D-22-24361R1Possible factors determining spatial patterns of drought impact on major crop yield sensitivity on a global scalePLOS ONE

Dear Dr. Hendrawan,

Thank you for submitting your manuscript to PLOS ONE. After careful consideration, we feel that it has merit but does not fully meet PLOS ONE’s publication criteria as it currently stands. Therefore, we invite you to submit a revised version of the manuscript that addresses the points raised during the review process.

We look forward to receiving your revised manuscript.

Kind regards,

Arun Kumar Shanker

Academic Editor

PLOS ONE

Reviewers' comments:

Reviewer's Responses to Questions

**Comments to the Author**

1. If the authors have adequately addressed your comments raised in a previous round of review and you feel that this manuscript is now acceptable for publication, you may indicate that here to bypass the “Comments to the Author” section, enter your conflict of interest statement in the “Confidential to Editor” section, and submit your "Accept" recommendation.

Reviewer #2: All comments have been addressed

Reviewer #3: (No Response)

2. Is the manuscript technically sound, and do the data support the conclusions?

Reviewer #2: Yes

Reviewer #3: (No Response)

3. Has the statistical analysis been performed appropriately and rigorously? 

Reviewer #2: Yes

Reviewer #3: (No Response)

4. Have the authors made all data underlying the findings in their manuscript fully available?

Reviewer #2: Yes

Reviewer #3: (No Response)

5. Is the manuscript presented in an intelligible fashion and written in standard English?

Reviewer #2: Yes

Reviewer #3: (No Response)

6. Review Comments to the Author

Reviewer #2: (No Response)

Reviewer #3: Authors suggested a study that measures the crop yield sensitivity based on machine learning approach. However, several issues are founded during review of the presented study such as:

• Revise the title in more meaningful style and avoid using many prepositions terms. Furthermore, consider the relation of your title with study contribution.

• Related works section is missing therefore how authors can justify they have solve very problematic issue that not resolved by existing studies? What is the main gap of the study?

• The contribution of the work is not clear for me there is no new model has presented. The contribution is basically is analysis several factors. Therefore, authors are suggested to highlight the contributions of the study in term of list at the end of related work section.

• Since authors used machine learning model for their analysis. An evaluation process should be conducted to check efficiency of used model for instance mean square error value. Another question why author used random forest model? Why not another model? Also the results from random forest should be compared with other machine learning model to show for the reader how far the selected model is better.

7. PLOS authors have the option to publish the peer review history of their article (what does this mean?). If published, this will include your full peer review and any attached files.

Reviewer #2: **Yes: **Emmanuel Amponsah Adjei

Reviewer #3: No

---

## [Author Response · Author response to Decision Letter 1]

5 Dec 2022

Please find our respond to reviewers attached.

---

## [Editor Report · Decision Letter 2]

3 Jan 2023

PONE-D-22-24361R2Possible factors determining global-scale patterns of crop yield sensitivity to droughtPLOS ONE

Dear Dr. Hendrawan,

Thank you for submitting your manuscript to PLOS ONE. After careful consideration, we feel that it has merit but does not fully meet PLOS ONE’s publication criteria as it currently stands. Therefore, we invite you to submit a revised version of the manuscript that addresses the points raised during the review process.

We look forward to receiving your revised manuscript.

Kind regards,

Arun Kumar Shanker

Academic Editor

PLOS ONE

Journal Requirements:

Additional Editor Comments :

The reviewers have suggested revisions

---

## [Author Response · Author response to Decision Letter 2]

3 Jan 2023

Dear Reviewers, 

We have addressed the minor issue raised by the editor as follows.

Journal Requirements:

We have reviewed the reference list and confirmed that it is complete and correct. 

We also did not cite papers that have been retracted. 

In addition, we have attached the Reference checking provided by MaRS editorial manager, 

showing that most citations are validated in CrossRef and Scopus.

Regarding additional Editor Comments: "The reviewers have suggested revisions."

We no longer find any reviewers' comments. 

Thank you for your time and contribution to improving our manuscript significantly.

Sincerely,

On behalf of the authors

---

## [Editor Report · Decision Letter 3]

20 Jan 2023

Possible factors determining global-scale patterns of crop yield sensitivity to drought

PONE-D-22-24361R3

Dear Dr. Hendrawan

We’re pleased to inform you that your manuscript has been judged scientifically suitable for publication and will be formally accepted for publication once it meets all outstanding technical requirements.

Kind regards,

Arun Kumar Shanker

Academic Editor

PLOS ONE

Additional Editor Comments (optional):

The reviwers have suggested minor revision
---

## [Editor Report · Acceptance letter]

24 Jan 2023

PONE-D-22-24361R3 

Possible factors determining global-scale patterns of crop yield sensitivity to drought 

Dear Dr. Hendrawan:

I'm pleased to inform you that your manuscript has been deemed suitable for publication in PLOS ONE. Congratulations! Your manuscript is now with our production department. 

Kind regards, 

on behalf of

Dr. Arun Kumar Shanker 

Academic Editor

PLOS ONE